# BiVO$_4$ Photoanodes Modified with Synergetic Effects between Heterojunction Functionalized FeCoO$_x$ and Plasma Au Nanoparticles

Huangzhaoxiang Chen [1], Qian Zhang [1,*], Aumber Abbas [2], Wenran Zhang [1], Shuzhou Huang [1], Xiangguo Li [1], Shenghua Liu [1,*] and Jing Shuai [1]

1 School of Materials, Shenzhen Campus, Sun Yat-Sen University, No. 66, Gongchang Road, Guangming District, Shenzhen 518107, China; chenhzhx@mail2.sysu.edu.cn (H.C.); zhangwr28@mail2.sysu.edu.cn (W.Z.); huangshzh6@mail2.sysu.edu.cn (S.H.); lixguo@mail.sysu.edu.cn (X.L.); shuaij3@mail.sysu.edu.cn (J.S.)
2 Songshan Lake Materials Laboratory, Room 425, C1 Building, University Innovation City, Songshan Lake, Dongguan 523000, China; aumber.abbas@sslab.org.cn
* Correspondence: zhangqian6@mail.sysu.edu.cn (Q.Z.); liushengh@mail.sysu.edu.cn (S.L.)

**Abstract:** The design and development of high-performance photoanodes are the key to efficient photoelectrochemical (PEC) water splitting. Based on the carrier transfer characteristics and localized surface plasmon resonance effect of noble metals, gold nanoparticles (AuNPs) have been used to improve the performance of photoanodes. In this study, a novel efficient composite BiVO$_4$/Au/FeCoO$_x$ photoanode is constructed, and the quantitative analysis of its performance is systematically conducted. The results reveal that the co-modification of AuNPs and FeCoO$_x$ plays a synergetic role in enhancing the absorption of ultraviolet and visible light of BiVO$_4$, which is mainly attributed to the localized surface plasmon resonance effect induced by AuNPs and the extended light absorption edge position induced by the BiVO$_4$/FeCoO$_x$ heterojunction. The BiVO$_4$/Au/FeCoO$_x$ photoanode exhibits a high photocurrent density of 4.11 mA cm$^{-2}$ at 1.23 V versus RHE at room temperature under AM 1.5 G illumination, which corresponds to a 299% increase compared to a pristine BiVO$_4$ photoanode. These results provide practical support for the design and preparation of PEC photoanodes decorated with AuNPs and FeCoO$_x$.

**Keywords:** solar water splitting; plasmon resonance effect; nanomaterials; heterojunction; photoelectrochemical





## 1. Introduction

Since the discovery of photoelectrochemical (PEC) water splitting based on semiconductor electrodes, scientists have devoted substantial efforts to transforming solar energy into clean and carbon-neutral hydrogen (H$_2$) energy in a stable, cost-effective, and efficient way [1]. The overall PEC water splitting consists of three main stages: (I) the absorption of light and the generation of electron–hole pairs in the semiconductor; (II) the separation of the electron and hole and their transfer to the surface of the semiconductor; (III) the occurrence of an oxygen evolution reaction (OER) on the photoanode and a hydrogen evolution reaction (HER) on the photocathode [2]. Due to the fracture of an O–H bond and the generation of an O–O bond, the oxygen evolution reaction on the surface of the photoanode is dynamically sluggish [3]. Thus, discovering and developing efficient photoanodes has always been regarded as the key to photoelectrochemical water splitting. In the past few decades, a wide variety of semiconductors, such as TiO$_2$ [4], Fe$_2$O$_3$ [5], and BiVO$_4$ [6], have been studied and employed as photoanodes.

Among the available photoanodes, BiVO$_4$ is undoubtedly an appealing photocatalyst material. It is an n-type semiconductor with a relatively narrow band gap of about 2.4 eV,

which enables it to absorb ultraviolet light and a wide range of visible light [7,8]. In addition, it has a suitable conduction band edge position which is close to the thermodynamic precipitation potential of $O_2$ [9,10]. Moreover, it is nontoxic and has impressive PEC performance in solution [11,12]. However, the slow carrier dynamics result in the recombination of bulk and interface charges of the unmodified $BiVO_4$, which limits its solar-to-hydrogen conversion efficiency. It has been widely verified that the actual maximum photocurrent density under AM 1.5 G illumination (100 mW cm$^{-2}$) of unmodified $BiVO_4$ is far below the theoretical value (7.5 mA cm$^{-2}$) [13].

In order to boost the OER of photoanodes, it is an effective strategy to modify $BiVO_4$ with oxygen evolution cocatalysts (OECs), which can reduce electron–hole recombination and enhance the photochemical performance. OECs like $FeCoO_x$ [14], $Cu_2O$ [15], $NiOOH$ [16], and $FeOOH$ [17] have been utilized and experimented with in modifying $BiVO_4$ photoanodes. Among them, a new type of $FeCoO_x$ has attracted researchers' attention due to its effective suppression of electron–hole recombination on the surface of $BiVO_4$ photoanode [18,19]. According to theoretical and experimental studies, it is clear that the substitution of cations in $CoO_x$ with iron ions results in abundant oxygen vacancies and forms a heterojunction with $BiVO_4$ [20,21]. $FeCoO_x$ also decreases the carrier transfer resistance to a certain extent, which enables the transfer of more photogenerated holes in the valence band of $BiVO_4$ to the $FeCoO_x$ layer. This leads to the generation of more reactive hydroxyl radicals to participate in water oxidation reactions efficiently [22,23]. However, the recombination center of the electron–hole may emerge on the interface of the heterojunction, leading to a decrease in PEC performance.

To promote bulk charge separation and carrier transfer, a practical method is to insert an electron transport layer (such as Pt, Au, Ni, etc.) into the middle of the heterojunction. As a classic plasmon nanostructure, the gold nanoparticles (AuNPs) are able to confine light in the vicinity of the surface and generate hot electron flow, which has a great effect on the chemical and PEC performance of the metal–oxide interface [24]. Since the work function of $BiVO_4$ is greater than gold, a Schottky junction is naturally formed when there is direct contact between $BiVO_4$ and Au. The Schottky junction may reduce the charge recombination on the metal–semiconductor interface, promote carrier transfer, and adjust the electronic band structure. Thus, the lifetime of photogenerated electrons and holes ought to increase, resulting in more electrons and holes involved in the PEC process [25].

On the other hand, the impact of geometric factors such as the size and shape of AuNPs on photoelectrochemical water splitting has been experimentally verified. AuNPs present particular photoelectronic and electrochemical properties when the particle size is adequately small (quantum size effect), such as localized surface plasmon resonance (LSPR) characteristics and electrochemical catalytic properties with great dependence on particle size [26,27]. Under the illumination of visible light or infrared light, the energy of oscillating electrons and the localized electromagnetic field generated by LSPR will be transferred to the conduction band of $BiVO_4$ via direct electron transfer (DET) or plasmon resonant energy transfer (PRET) [10,28,29]. In the case of $BiVO_4$/Au, numerous experiments indicate that energy transfers via DET, implying that the excited hot electron of Au may overcome the metal–semiconductor barrier and transfer to the conduction band of $BiVO_4$, contributing to improved photocurrent density.

Herein, in order to progressively enhance the PEC properties of $BiVO_4$ photoanodes and verify the synergism of AuNPs and $FeCoO_x$, the $BiVO_4$/Au/$FeCoO_x$ photoanode is constructed through a template route, which is followed by electrochemical deposition of AuNPs and $FeCoO_x$, respectively. The optimized $BiVO_4$/Au/$FeCoO_x$ photoanode exhibits a high photocurrent density of 4.11 mA cm$^{-2}$ at 1.23 V versus RHE under AM 1.5 G (100 mW cm$^{-2}$) illumination, which is over three-fold that of the pristine $BiVO_4$ photoanode and a 29.7% increase in that of the $BiVO_4$/$FeCoO_x$ counterpart.

## 2. Results and Discussion

Figure 1a displays a schematic diagram of the preparation process for the $BiVO_4/Au/FeCoO_x$ photoanode, and the experimental details are presented in the experimental section. The $BiVO_4/Au/FeCoO_x$ photoanode X-ray diffraction (XRD) patterns are shown in Figure 1b. The characteristic peaks of crystal planes such as (020), (011), and (121) of $BiVO_4$ have been labeled. The peaks corresponding to the (111) and (200) planes of Au have also been marked, but due to the small content and size, more diffraction peaks of AuNPs and $FeCoO_x$ are too difficult to detect. Figure 1c displays the scanning electron microscopy (SEM) image of the $BiVO_4/Au/FeCoO_x$ photoanode, which implies that the co-modification of AuNPs and $FeCoO_x$ makes the originally striped $BiVO_4$ thicker and more compact. The surface of the $BiVO_4/Au/FeCoO_x$ photoanode material is uniformly filled with a similar cluster structure. Figure S1a shows the SEM image of a pristine $BiVO_4$ photoanode. It can be seen that $BiVO_4$ is a thin film formed by particles, strips, or clusters, with particle sizes ranging from 100 nm to 300 nm, while the length of the strip formed by the particles is approximately 300 nm to 1 μm. Additionally, due to issues with the preparation methods, there are unfilled holes in $BiVO_4$. Figure S1b shows the SEM image of the $BiVO_4/Au$ photoanode, in which the particles with significantly brighter brightness than $BiVO_4$ are gold nanoparticles, with a particle size of approximately 80~150 nm. Figure S1c shows the SEM image of the $BiVO_4/FeCoO_x$ photoanode. It can be seen that $FeCoO_x$ adheres to the surface of $BiVO_4$ in a paste-like form, making the original strip or cluster structure of $BiVO_4$ more flexible.

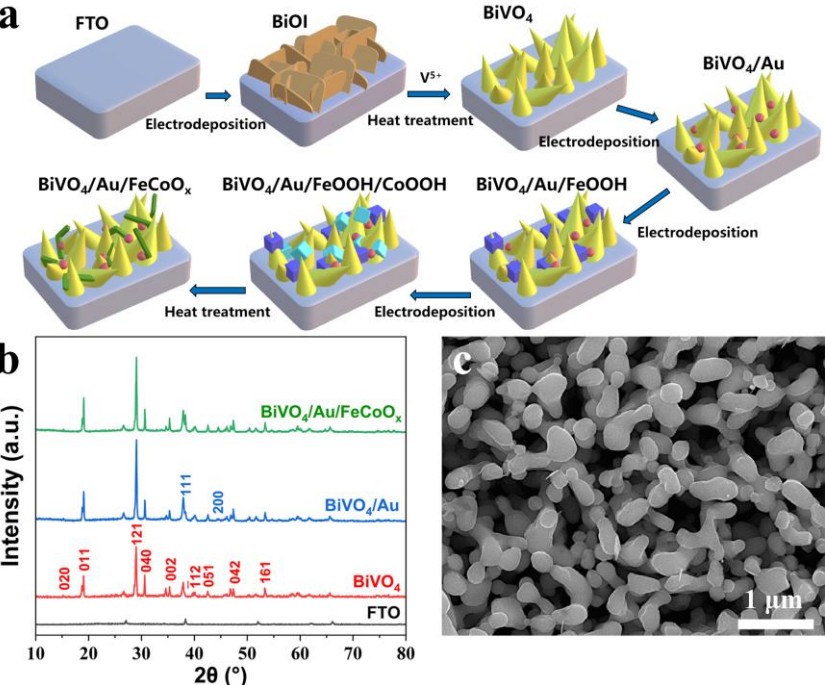

**Figure 1.** (**a**) Diagram for the preparation procedure of the $BiVO_4/Au/FeCoO_x$ photoanode. (**b**) XRD patterns of photoanodes. (**c**) SEM image of $BiVO_4/Au/FeCoO_x$.

Figure 2a–g display the energy dispersive spectroscopy (EDS) spectrum of the $BiVO_4/Au/FeCoO_x$ composite photoanode. The distribution of six elements, Bi, V, O, Au, Fe, and Co, can be seen from the EDS elemental mapping, which can also prove the existence of these elements. From the mapping of Au in Figure 2e, it can be seen that there are three local positions with dense Au distribution in the upper left, upper right, and lower middle directions, which may be caused by large-size AuNPs. As shown in Figure 2h and Table S1, the content information of each element in the selected area indicates that the contents of Au, Fe, and Co are very low among the six kinds of elements and appear to

have atomic ratios of 1.11%, 1.28%, and 0.58%, respectively, indicating that Au and FeCoO$_x$ mainly play a modifying role in the photoanodes. Moreover, it can also be calculated that the atomic ratio of Fe and Co in the selected area is Fe:Co = 2.2:1. In the relevant publications on the preparation of Au and FeCoO$_x$, the content of Au, Fe, and Co is generally below 5% [30], and the most significant improvement in the PEC performance of FeCoO$_x$ at present belongs to the work of Wang et al., where the relative atomic ratio of Fe:Co is Fe:Co = 1.1:1 [14].

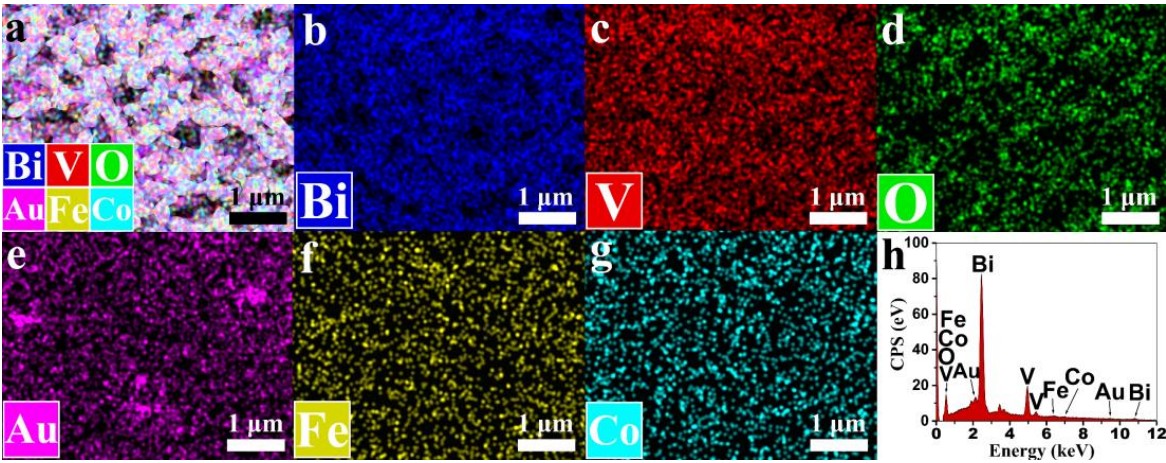

**Figure 2.** Elemental mapping of BiVO$_4$/Au/FeCoO$_x$. (**a**) Overall image. (**b**) Bi. (**c**) V. (**d**) O. (**e**) Au. (**f**) Fe. (**g**) Co. (**h**) EDS energy distribution.

Figure 3 displays the high-resolution X-ray photoelectron spectroscopy (XPS) elemental spectrum of the BiVO$_4$/Au/FeCoO$_x$ photoanode, which contains information on the composition and chemical states of elements in the composite photoanode. The high-resolution Bi 4f spectrum can be fitted by two main peaks, located at 159.1l eV and 164.41 eV, respectively, corresponding to the binding energy of the Bi 4f$_{7/2}$ spin state and the Bi 4f$_{5/2}$ spin state. The high-resolution V 2p spectrum displays two main peaks at 516.70 eV and 524.16 eV, respectively, and they correspond to the binding energy of the V 2p$_{3/2}$ spin state and the V 2p$_{1/2}$ spin state. Figure 3c displays the high-resolution O 1s spectrum, fitted with three main peaks located at 530.45 eV, 531.55 eV, and 532.35 eV, respectively. These three peaks correspond to O$^{2-}$ in the lattice (O$_L$), hydroxyl bound to metal cations (O$_V$), and oxygen in the chemically adsorbed or dissociated state (such as dissociated CO$_3$$^{2-}$, absorbed H$_2$O, absorbed O$_2$, etc., O$_C$). From Figure 3c, it can be seen that the O$_L$ peak is quite prominent compared with the O$_V$ and O$_C$ peaks, reflecting that the oxygen element is mainly present in the form of O$^{2-}$ in the lattice. The existence of the O$_V$ peak proves the presence of a large number of oxygen defects or vacancies on the surface of the BiVO$_4$/Au/FeCoO$_x$ photoanode, while the existence of the O$_C$ peak proves the presence of surface hydroxyl groups on the surface of the BiVO$_4$/Au/FeCoO$_x$ photoanode. The presence of surface hydroxyl groups can promote the transport and capture of photogenerated electrons and photogenerated holes, thereby reducing the recombination of photogenerated carriers and improving the photoelectric conversion efficiency of BiVO$_4$/Au/FeCoO$_x$. The high-resolution Au 4f spectrum displays two main peaks, 83.82 eV and 87.51 eV, respectively, corresponding to the binding energy of the Au 4f$_{7/2}$ spin state and the Au 4f$_{5/2}$ spin state. The high-resolution Fe 2p spectrum is fitted with six peaks, including three main peaks and three satellite peaks. The three main peaks are located at 710.31 eV, 712.42 eV, and 723.97 eV. Among them, the peak located at 723.97 eV corresponds to the Fe 2p$_{1/2}$ peak, and the first two of the three main peaks represent Fe$^{2+}$ and Fe$^{3+}$, respectively, which together form the Fe 2p$_{3/2}$ peak. The atomic ratio of these two peaks can also be calculated from the peak position table as Fe$^{3+}$:Fe$^{2+}$ = 1.89:1. The high-resolution Co 2p spectrum consists of six fitted peaks, including three main peaks and three satellite peaks. The

three main peaks are located at 779.60 eV, 780.60 eV, and 795.74 eV, where the peak at 795.74 eV corresponds to the Co $2p_{1/2}$ peak. The first two of the three main peaks represent $Co^{3+}$ and $Co^{2+}$, respectively, and together, they form the Co $2p_{3/2}$ peak. The atomic ratio of these two ions can be calculated from the peak position table shown in Table S2, as $Co^{2+}:Co^{3+} = 3.59:1$. The atomic ratio of Fe and Co can also be calculated as Fe:Co = 1.37:1.

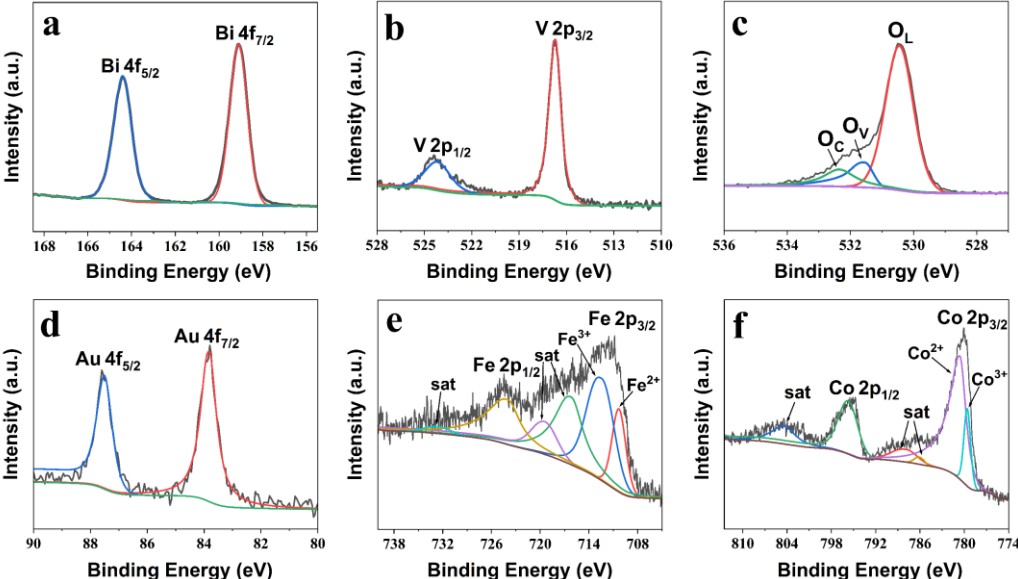

**Figure 3.** High-resolution XPS spectrum of the $BiVO_4/Au/FeCoO_x$. (**a**) Bi 4f, (**b**) V 2p, (**c**) O 1s, (**d**) Fe 2p, (**e**) Co 2p, and (**f**) Au 4f.

Figure 4a displays the ultraviolet–visible (UV-vis) absorption spectra of $BiVO_4$, $BiVO_4/Au$, and $BiVO_4/Au/FeCoO_x$ photoanodes. The FTO substrate has been used as the background to eliminate the influence of FTO on the absorbance. It can be seen that the light absorption edge of $BiVO_4$ is around 510 nm, which is consistent with its band gap of 2.3~2.5 eV. The light absorption edge of $BiVO_4/Au$ remains almost unchanged, but its absorption wavelength range from 300 nm to 900 nm is higher than that of pristine $BiVO_4$. The light absorption edge of $BiVO_4/Au/FeCoO_x$ photoanodes is about 660 nm, and the light absorption at wavelengths of 300~900 nm is stronger than that of $BiVO_4$ and $BiVO_4/Au$. The loading of AuNPs and $FeCoO_x$ has played a synergic role in enhancing the absorption of UV-vis light of $BiVO_4$, which is mainly due to the LSPR effect induced by AuNPs and the extended light absorption edge position induced by the $BiVO_4/FeCoO_x$ heterojunction.

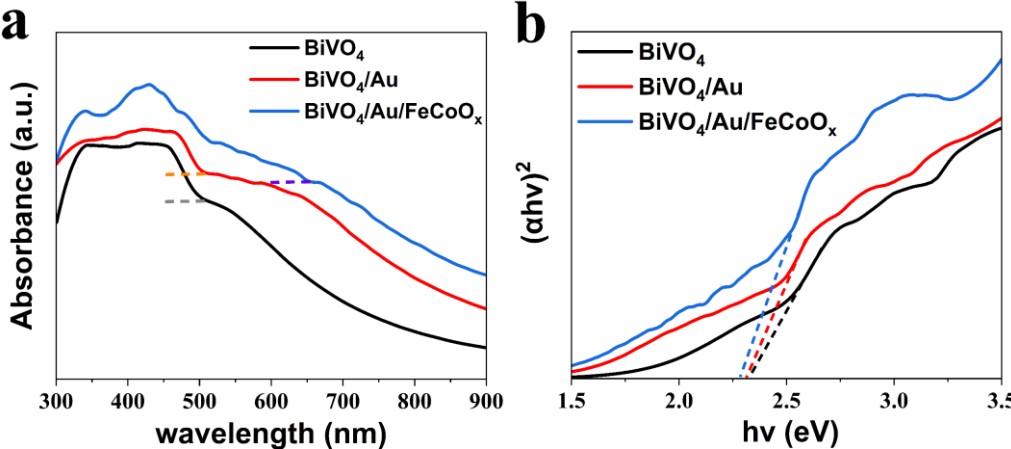

**Figure 4.** Light absorbance of photoanodes. (**a**) UV-vis diffusion absorption spectra. (**b**) Tauc plots of $BiVO_4$ films modified with cocatalysts.

The band gap can also be obtained by Tauc plots. Since most of the valence band electrons and conduction band electrons of the semiconductor are distributed near the band gap, when the energy of a photon is close to the band gap, a large number of electrons can be excited from the top of the valence band to the bottom of the conduction band by absorbing the photon energy. At this moment, the absorption coefficient of the semiconductor will increase with the increase in the number of photons. The relationship between the band gap and the absorption coefficient of the semiconductor materials can be expressed as follows:

$$(\alpha h v)^n = A * (h v - E_g) \tag{1}$$

where $\alpha$ is the absorption coefficient, $h \approx 4.13567 \times 10^{-15}$ eV s, as Planck's coefficient, $v$ is the frequency of the incident photon, $A$ is the proportional coefficient, and $E_g$ is the band gap of the semiconductor. The value of n is related to the type of semiconductor. When the semiconductor is a direct band gap semiconductor, that is, the conduction band bottom and valence band top of the semiconductor correspond to the same wave vector of electronic states, a direct band edge transition can occur, where n = 2. When the band gap of a semiconductor is an indirect band gap, that is, the conduction band bottom and valence band top of the semiconductor are not at the same wave vector of electronic states, the band edge transition requires the participation of phonons, where n = 1/2 at this time. Since $BiVO_4$ is a direct band gap semiconductor, n is set as 2. Substitute the absorption coefficient and others into Equation (1) and create an $(\alpha h v)^2 - h v$ diagram. As shown in Figure 4b, the band gap of $BiVO_4$ and $BiVO_4/Au$ is about 2.32 eV, while the band gap of $BiVO_4/Au/FeCoO_x$ is about 2.28 eV. These results indicate that the modification of AuNPs has little effect on the band gap of $BiVO_4$, while the band gap narrowing of $BiVO_4/Au/FeCoO_x$ photoanodes can be attributed to the narrow band gap of $FeCoO_x$ (2.04 eV) compared to $BiVO_4$ (2.3~2.5 eV).

Figure 5a displays the linear sweep voltammetry (LSV) curve of the photoanodes. It can be seen that the photocurrent density under dark conditions is only $7 \times 10^{-7}$ mA cm$^{-2}$ at 1.23 V vs. RHE, and the magnitude of this photocurrent density can be regarded as almost no photocurrent generation. Under AM 1.5 G illumination at 1.23 V vs. RHE, the photocurrent density of $BiVO_4/Au/FeCoO_x$ reached 4.11 mA cm$^{-2}$, which is nearly four times than that of pristine $BiVO_4$ (1.03 mA cm$^{-2}$). This photocurrent density is increased by 71.8% compared to $BiVO_4/Au$ (1.77 mA cm$^{-2}$) and increased by 29.7% compared to $BiVO_4/FeCoO_x$ (3.17 mA cm$^{-2}$). Additionally, the $BiVO_4/Au/FeCoO_x$ photoanode exhibits an onset potential of 0.245 V vs. RHE, which is the lowest among the photoanodes and cathodically shifted by about 27 mV compared with that of pristine $BiVO_4$. There are multiple reasons for this increase in the photocurrent density. For example, according to the discussions of Figure 4a, the modification of $FeCoO_x$ has expanded the original light absorption edge of $BiVO_4$ from only about 510 nm to about 660 nm. On the other hand, the modification of AuNPs and $FeCoO_x$ has played a positive role in improving the absorbance of $BiVO_4$ for ultraviolet and visible light, which is consistent with the results of UV-vis spectra. It is not difficult to understand that under the assumption of the same photoelectric conversion efficiency, the stronger the absorption of light by the photoanode, the greater the photocurrent density generated. Moreover, the synergism effect of AuNPs and $FeCoO_x$ may contribute to the enhancement of the photocurrent density. For example, the photogenerated holes in the valence band of $BiVO_4$ will be captured in the $FeCoO_x$ surface and participate in the PEC water oxidation, while the excited hot electron of AuNPs may overcome the metal–semiconductor barrier and transfer to the conduction band of $BiVO_4$. Namely, the quality of $FeCoO_x$ as an oxygen evolution cocatalyst is of vital significance in promoting PEC water oxidation, which may cause differences in the photocurrent between photoanode counterparts modified with $FeCoO_x$ [14].

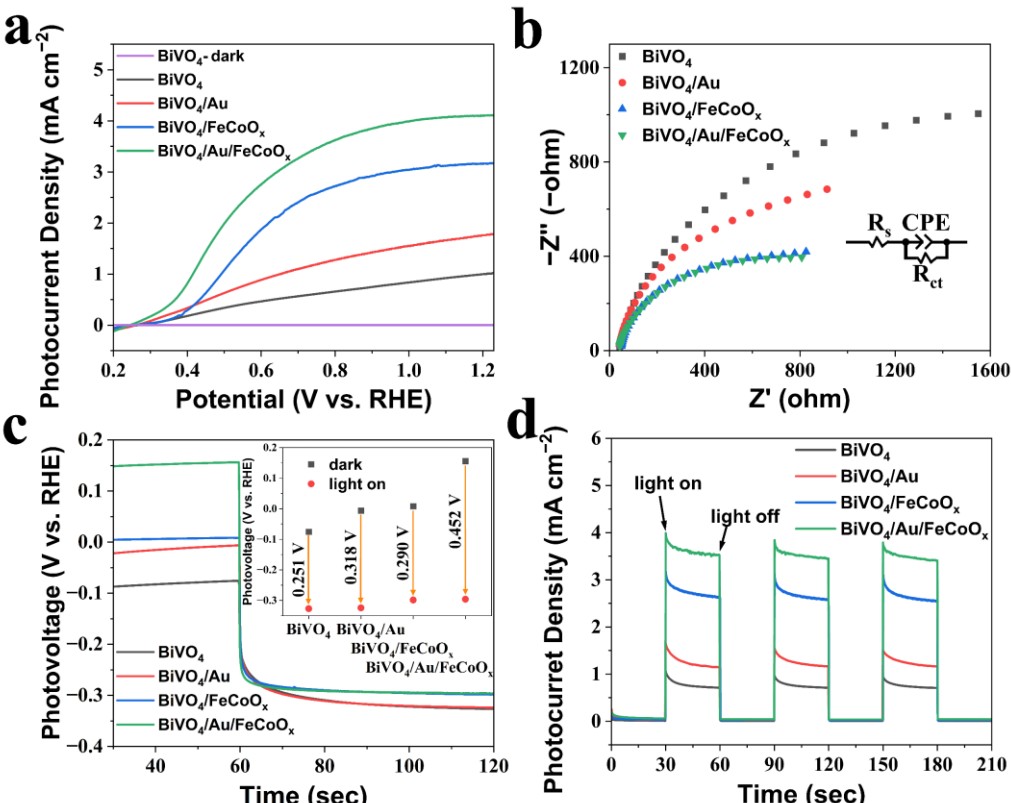

**Figure 5.** PEC performance of photoanodes in 0.5 M potassium phosphate buffer (pH = 7.0) AM 1.5 G (100 mW cm$^{-2}$). (**a**) LSV curves. (**b**) EIS curves. (**c**) Photovoltage curves. (**d**) j-t curve of the photoanodes at 1.23 V vs. RHE under intermittent illumination.

The increase in photocurrent density can also be attributed to a decrease in surface charge recombination. Figure 5b represents the electrochemical impedance spectroscopy (EIS) of the prepared $BiVO_4$, $BiVO_4/Au$, $BiVO_4/FeCoO_x$, and $BiVO_4/Au/FeCoO_x$ photoanodes. The EIS spectra can be explained with the Randles equivalent circuit model, which contains two basic elements, $R_s$ and $R_{ct}$ [31]. The element $R_s$ is the resistance relating to carrier transfer, containing the resistance of FTO substrates, the electrolyte, and wire connections in the circuit. The diameter of the circle corresponding to the semicircle or arc in the middle-frequency region of EIS reflects the interface resistance at the photoelectrode–electrolyte interface, also known as the elements of $R_{ct}$ or charge transfer resistance. Figure 5b mainly captures the arced part of the middle-frequency region. It can be easily seen from Figure 5b that the modification of AuNPs and $FeCoO_x$, respectively, reduced the diameters of the $BiVO_4$ photoanodes to some extent, while the diameter of the Nyquist plot of $BiVO_4/Au/FeCoO_x$ is the lowest among the photoanodes. These results indicate that the co-modification of photoanode with AuNPs and $FeCoO_x$ plays a positive role in reducing interface resistance and the recombination of photogenerated electrons and holes on the photoanode-electrolyte interface synergically, thereby promoting the participation of photogenerated holes in the water oxidation reaction and increasing the photocurrent density. Figure 5c displays the photovoltage changes of the photoanodes with or without illumination. Theoretically, the photovoltage is equal to the difference between the open circuit voltage under light and dark conditions. Compared to the pristine $BiVO_4$ with a photovoltage of 0.251 V, the photovoltage of $BiVO_4/Au$ increases to 0.318 V. The photovoltage of $BiVO_4/FeCoO_x$ increases to 0.290 V, and the photovoltage of $BiVO_4/Au/FeCoO_x$ increases to 0.452 V, which is 80.1% higher than the pristine $BiVO_4$. The photovoltage results are in agreement with the onset potential of photoanodes shown in LSV curves in Figure 5a. It can be concluded that the loading of AuNPs and $FeCoO_x$, to some extent, increased the photovoltage of the photoanodes. This results in the cathodic on-

set potential shifting, indicating that the synergism of AuNPs and $FeCoO_x$ not only boosts OER kinetics on the photoanodes but also contributes to thermodynamics, providing a greater driving force on PEC water splitting [32].

Figure 5d displays the j–t curve of the photoanodes at 1.23 V vs. RHE and intermittent AM 1.5 G illumination (100 mW cm$^{-2}$), with periods of 30~60 s, 90~120 s, and 150~180 s. It can be seen that when the condition first enters the illumination from the dark, each photoanode exhibits a peak of decreased photocurrent density, and the sharpness of this peak reflects the degree of surface charge recombination. Therefore, it can be said that the surface charge recombination on each photoanode is still quite severe.

Figure S2 displays the applied bias photon-to-current efficiency (ABPE) diagram of the photoanodes by substituting the LSV curve of the photoanode material into Equation (3) with a Faraday efficiency of 100% and a $P_{in}$ of 100 mW cm$^{-2}$. From Figure S2, it can be seen that $BiVO_4$ reached a peak ABPE of 0.31% at a potential of 0.69 V vs. RHE, while $BiVO_4$/Au reached a peak of 0.57% at about 0.68 V vs. RHE. $BiVO_4$/$FeCoO_x$ reached a peak of 1.32% at 0.70 V vs. RHE, and the target photoanode $BiVO_4$/Au/$FeCoO_x$ reached a peak ABPE of 1.75% at about 0.64 V vs. RHE, which is more than five times the peak value of pristine $BiVO_4$. These data are sufficient to demonstrate the significant impact of AuNP modification on improving photoelectric conversion efficiency and reducing surface charge recombination, among which, the most attractive aspect for researchers is the significant increase in photocurrent density.

Figure S3 displays the durability test of the photoanodes at 1.23 V vs. RHE under AM 1.5 G. It can be seen that within about 30 min after the start of durability testing, the photocurrent density of each photoelectrochemical system decreased rapidly. In this period, the photocurrent density of $BiVO_4$/Au/$FeCoO_x$ decreased by 56.3% from the initial value of 3.98 mA cm$^{-2}$ to 1.75 mA cm$^{-2}$. The photocurrent density of $BiVO_4$/$FeCoO_x$ decreased from the initial 3.07 mA cm$^{-2}$ to 1.25 mA cm$^{-2}$, a relative decrease of 59.3%. The photocurrent density of $BiVO_4$ decreased from the initial 1.08 mA cm$^{-2}$ to 0.64 mA cm$^{-2}$, a relative decrease of 40.7%. These results indicate that there is still room for improvement in the quality of $FeCoO_x$ and AuNPs, while they may quickly dissolve in the electrolyte over time, which is consistent with the results of the photocurrent density compared with works in other publications. Namely, the stability of $BiVO_4$/Au/$FeCoO_x$ has decreased to a certain extent, which is one of the effects of the modification of AuNPs and $FeCoO_x$. However, compared to the weakened stability of $BiVO_4$/Au/$FeCoO_x$, the beneficial effect of increasing photocurrent density is more significant.

The schematic diagram of a photoelectrochemical system using $BiVO_4$/Au/$FeCoO_x$ as a composite photoanode is shown in Figure 6. The following is the proposed working mechanism of the $BiVO_4$/Au/$FeCoO_x$ composite photoanode under illumination in phosphoric acid buffer, which consists of five processes: (I) electrons on AuNPs are excited by LSPR under incident light irradiation, (II) photogenerated hot electrons of AuNPs are transferred to the conduction band of $BiVO_4$ and collected by external circuits, (III) photogenerated holes on the valence band of $BiVO_4$ can be extracted and stored in the $FeCoO_x$ layer, and photogenerated electrons are extracted and transferred to the AuNPs, (IV) water is oxidized by holes on $FeCoO_x$, leading to the formation of oxygen and the elimination of holes, and (V) electron-deficient AuNPs are reduced by photogenerated electrons from $BiVO_4$, returning to their original metal state.

Due to the fact that the work function of $BiVO_4$ is different from Au, this difference in work function causes the energy bands of $BiVO_4$ to bend, which naturally forms a Schottky junction or Schottky barrier. Under illumination, the photogenerated hot electrons of Au can be amplified by LSPR, therefore becoming more likely to flow across the Schottky barrier into the conduction band of $BiVO_4$ and be collected by external circuits [24]. Thus, the electrons transmitted from AuNPs to $BiVO_4$ in process II are reasonable. In addition, the introduction of AuNPs and $FeCoO_x$ plays a synergic role in the photoelectrocatalysis kinetics and thermodynamics. The promoted charge transfer process will reduce the recombination of photogenerated electron–hole pairs on the electrode–electrolyte interface [28,33].

Therefore, compared to the pristine $BiVO_4$, the lifetime of photogenerated electrons and photogenerated holes is expected to improve. More electrons and holes participate in the PEC water splitting, indicating that the charge utilization rate will also be improved. Meanwhile, shown in Figure 6 as a hole storage layer, $FeCoO_x$ also forms a functionalized p-n junction with $BiVO_4$ and boosts the photogenerated holes in the valence band of $BiVO_4$ to be captured and stored in the $FeCoO_x$ surface. The stored holes can further participate in the effective water oxidation reaction to form reactive hydroxyl radicals. Thus, $FeCoO_x$ cocatalysts promote the catalytic kinetics in the electrode–electrolyte interface. Based on the above analysis, it is believed that AuNPs and $FeCoO_x$ can exert a synergistic effect on the PEC performance improvement of the $BiVO_4$.

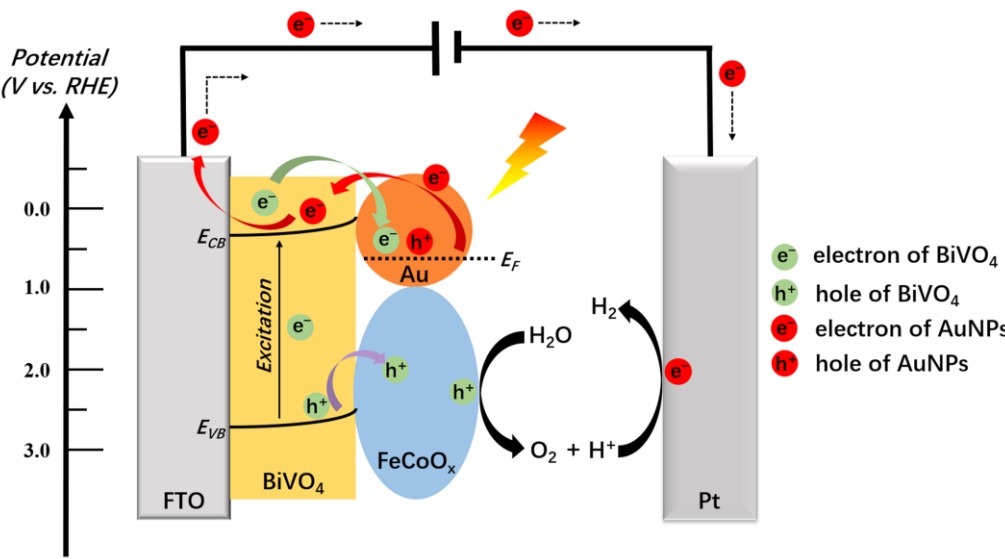

**Figure 6.** Proposed mechanism of the $BiVO_4/Au/FeCoO_x$ photoanode for PEC water splitting.

### 3. Materials and Methods

#### 3.1. Preparation of $BiVO_4$ Films

Ethanol solution with 0.23 M 1,4-benzoquinone was mixed with water solution with 0.4 M Bi $(NO_3)_3$ and 0.4 M KI in a ratio of 7:17 to prepare the electrolyte, and the pH was adjusted to 1.70 with nitric acid. Ag/AgCl was used as the reference electrode, platinum electrode as the counter electrode, and FTO glass as the working electrode. The bismuth precursor film BiOI was deposited on FTO glass for 300 s at −0.1 V. Dimethyl sulfoxide (DMSO) dissolved in 0.2 M acetylacetone vanadium oxide was dropped onto BiOI and kept in air at 450 °C for 2 h at a heating rate of approximately 2 °C/min. Finally, BiOI was soaked in 1 M NaOH for 30 min to remove $V_2O_5$ and was rinsed with pure water.

#### 3.2. Preparation of AuNPs

Briefly, 0.005 M chloroauric acid solution was used as the electrolyte, Ag/AgCl was the reference electrode, and a platinum electrode was the counter electrode. AuNPs were deposited at 0.15 V for 40 s. Then, they were rinsed with pure water.

#### 3.3. Preparation of $FeCoO_x$ Cocatalysts

The $FeCoO_x$ cocatalyst was deposited through two steps under AM 1.5 G. Briefly, 0.1 M $FeSO_4$ solution was used as the electrolyte, while Ag/AgCl was used as the reference electrode, and platinum electrode as the counter electrode. The FeOOH layer was deposited at 0.25 V for 300 s. Then, 0.025 M $(CH_3COO)_2Co$ solution was used as the electrolyte, platinum electrode as the counter electrode, and Ag/AgCl as the reference electrode, and $CoO_x$ was deposited at 0.25 V for 40 s. Finally, the $FeCoO_x$ cocatalyst was obtained after keeping it in air at 500 °C for 2 h

### 3.4. Characterization

The morphologies and structures of the photoanodes were characterized using SEM (JSM-IT500, JEOL) coupled with EDS, and XRD (D8 Advance, Bruker) with Cu K$\alpha$ radiation, respectively. The composition and chemical states of photoanodes were characterized by XPS (ESCALAB xi+). UV-vis absorption spectra were recorded by an Evolution 220 spectrophotometer.

### 3.5. PEC measurements

The PEC measurements were completed through electrochemical workstations and xenon lamps. The test circuit was constructed under AM 1.5 G illumination using a classic three-electrode system (platinum as the counter electrode, Ag/AgCl as the reference electrode) and 0.5 M phosphoric acid buffer (pH = 7.00). Photocurrent density potential curves were obtained using linear sweep voltammetry ranging from approximately 0.1 to 1.4 V vs. RHE with a scanning rate of 0.01 V s$^{-1}$. Electrochemical impedance spectra were obtained at open circuit potential with a frequency ranging from roughly 100 kHz to 0.01 Hz and a signal amplitude of 0.01 V. Photovoltage was measured at open circuit potential under AM 1.5 G for 60 s and then continued for 60 s in the dark condition.

All the potentials vs. RHE can be converted from the potentials vs. Ag/AgCl following

$$E_{RHE} = E_{Ag/AgCl} + E^0_{Ag/AgCl} + 0.059 * pH \tag{2}$$

where $E_{RHE}$ is converted potential vs. RHE, $E^0{}_{Ag/AgCl}$ is 0.197 V at room temperature (25 °C), and $E_{Ag/AgCl}$ is the potential vs. Ag/AgCl.

*ABPE* can be calculated according to

$$ABPE = \frac{j * (1.23 - V_b)}{P_{in}} * 100\% \tag{3}$$

where $j$ is the photocurrent density (mA cm$^{-2}$), $V_b$ is the corresponding potential vs. RHE, and $P_{in}$ is the illumination intensity (100 mW cm$^{-2}$).

## 4. Conclusions

In summary, to progressively enhance the PEC properties of $BiVO_4$ photoanodes, a novel $BiVO_4/Au/FeCoO_x$ photoanode is designed and the synergetic role of AuNPs and $FeCoO_x$ is verified. The characterization, optical properties, and PEC performance of optimized photoanodes are thoroughly investigated. The results reveal that the co-modification of AuNPs and $FeCoO_x$ plays a synergic role in enhancing the absorption of ultraviolet and visible light of $BiVO_4$. This is mainly attributed to the LSPR effect induced by AuNPs and the extended light absorption edge position induced by the $BiVO_4/FeCoO_x$ heterojunction. Moreover, the appropriately sized Au nanoparticles and $FeCoO_x$ cocatalysts played a positive role in increasing the photovoltage of $BiVO_4$ and in reducing the charge transfer resistance on the electrode–electrolyte interface, consequently leading to the promotion of the separation and migration of photogenerated carriers and the prominent enhancement of the photocurrent density. This work provides practical experimental support and a theoretical explanation for the design of effective PEC photoanodes.

**Supplementary Materials:** The following supporting information can be downloaded at: https://www. mdpi.com/article/10.3390/catal13071063/s1, Figure S1: SEM images of (a) $BiVO_4$, (b) $BiVO_4/Au$, and (c) $BiVO_4/FeCoO_x$.; Figure S2: ABPE curves of photoanodes in 0.5 M potassium phosphate buffer (pH=7.0) under AM 1.5G (100 mW cm−2); Figure S3: Durability tests of photoanodes in 0.5 M potassium phosphate buffer (pH=7.0) under AM 1.5G (100 mW cm−2) at 1.23 V vs. RHE. Table S1: Content Information of Each Element in Mapping of $BiVO_4/Au/FeCoO_x$. Table S2: Peak position and atomic content in XPS results.

**Author Contributions:** Methodology, H.C. and Q.Z.; Project administration, J.S.; Resources, Q.Z. and S.L.; Writing original draft, H.C. and Q.Z.; Writing review and editing, H.C., Q.Z., A.A., S.H. and S.L. Data curation, X.L. and W.Z.; Formal analysis, S.L.; Investigation, J.S. All authors have read and agreed to the published version of the manuscript.

**Funding:** This research was funded by the Shenzhen Science and Techonlogy Program (Grant No. 202206193000001), Guangdong Basic and Applied Basic Research Foundation (Grant No. 2023A151501207, 22022A1515110619), the Open Project Fund from Guangdong Provincial Key Laboratory of Materials and Technology for Energy Conversion, Guangdong Technion-Israel Institute of Technology, Grant No.MATEC2023KF003.

**Data Availability Statement:** All data generated or analyzed during this study are included in this published article.

**Conflicts of Interest:** The authors declare no conflict of interest.

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
