# Peer review of "BiVO4 Photoanodes Modified with Synergetic Effects between Heterojunction Functionalized FeCoOx and Plasma Au Nanoparticles"

_catalysts, doi:10.3390/catal13071063_

Round 1

Reviewer 1 Report

Please find attached review response.

Reviewer 2 Report

The submitted article is devoted to the synthesis and study of BiVO4/Au/FeCoOx photoanodes for efficient solar water oxidation and is of interest to specialists in this field.

The paper is structured correctly; the abstract corresponds to the main content of the article. All the methods used in the work are carefully described. These methods are modern, reasonable, accurate and widely used in the research practice. The main key results obtained in the work are listed in the conclusion. The work can be accepted for publication after the revisions.

Comments on the work are given below.

1. Introduction, lines 45, 46. The toxicity of BiVO4 vanadium is not discussed in refs. 11, 12.

2. Please improve manuscript by comparing your results with those reported in [14] for the FeCoOx/BiVO4 photoanode. The methods for preparing these photoanodes in both works are practically the same, but the results are different.

3. There is no data on the presence of gold nanoparticles. SEM images of gold nanoparticles and their size should be given since the size of nanoparticles is a key parameter of their functionality. Unfortunately, without these results, the conclusions of the work are not fully justified.

Round 2

Reviewer 1 Report

1. The corrected pdf version of the text contains deleted and added text. For this reason, it is hard to follow manuscript.

2. Author should use current density instead of density at Fig. 5a.

3. Within the text authors have used Rohm and therefore electric equivalent circuit within Fig. 5b should contain Rohm instead of Rs. Also, within the text authors should discuss about Rcr, not about half of Rct. In order to analyse electrochemical impedance spectroscopy, it is important to use appropriate software (for example ZSimpWin). It is not acceptable to fit the semicircle in this way and to determine half of Rct (Table 3), as well as to discuss about half of Rct. If fitting software is not available, please compare resistance from Nyquist plot without fitting.

4. Authors did not comment Figure S6 in manuscript.

Reviewer 2 Report

The authors took into account most of the comments of the reviewers and revised the manuscript accordingly. From my point of view, the article is suitable for publication in the journal Catalysts.
